# Service user experiences of community services for complex emotional needs: A qualitative thematic synthesis

Luke Sheridan Rains[1]*, Athena Echave[2], Jessica Rees[2], Hannah Rachel Scott[2], Billie Lever Taylor[1], Eva Broeckelmann[3], Thomas Steare[1], Phoebe Barnett[4], Chris Cooper[4], Tamar Jeynes[3], Jessica Russell[3], Sian Oram[5], Sarah Rowe[2], Sonia Johnson[1,6]

1 Division of Psychiatry, NIHR Mental Health Policy Research Unit, University College London, London, United Kingdom, 2 Division of Psychiatry, University College London, London, United Kingdom, 3 Health Service and Population Research Department, NIHR Mental Health Policy Research Unit Complex Emotional Needs Lived Experience Working Group, Institute of Psychiatry, Psychology & Neuroscience, King's College London, London, United Kingdom, 4 Department of Clinical, Educational and Health Psychology, University College London, London, United Kingdom, 5 Health Service and Population Research Department, NIHR Mental Health Policy Research Unit, Institute of Psychiatry, Psychology & Neuroscience, King's College London, London, United Kingdom, 6 Camden and Islington NHS Foundation Trust, London, United Kingdom

* l.sheridanrains@ucl.ac.uk

## Abstract

### Background

There is a recognised need to develop clear service models and pathways to provide high quality care in the community for people with complex emotional needs, who may have been given a "personality disorder" diagnosis. Services should be informed by the views of people with these experiences.

### Aims

To identify and synthesise qualitative studies on service user experiences of community mental health care for Complex Emotional Needs.

### Methods

We searched six bibliographic databases for papers published since 2003. We included peer reviewed studies reporting data on service user experiences and views about good care from community-based mental health services for adults with CEN, including generic mental health services and specialist "personality disorder" services. Studies using any qualitative method were included and thematic synthesis used to identify over-arching themes.

### Results

Forty-seven papers were included. Main themes were: 1) The need for a long-term perspective on treatment journeys; 2) The need for individualised and holistic care; 3) Large variations in accessibility and quality of mental health services; 4) The centrality of therapeutic

**Data Availability Statement:** This is a systematic review. The data underlying the results presented in the study are available in published literature. Details of our methods are available in the paper. A

detailed search strategy and precise inclusion criteria are available in the supplementary information.

**Funding:** This paper presents independent research commissioned and funded by the National Institute for Health Research (NIHR) Policy Research Programme, conducted by the NIHR Policy Research Unit (PRU) in Mental Health. The views expressed are those of the authors and not necessarily those of the NIHR, the Department of Health and Social Care or its arm's 591 length bodies, or other government departments.

**Competing interests:** The authors have declared that no competing interests exist.

relationships; 5) Impacts of 'personality disorder' diagnosis. Themes tended to recur across studies from different countries and years.

## Discussion

Recurrent major themes included wanting support that is individualised and holistic, provides continuity over long journeys towards recovery, and that is delivered by empathetic and well-informed clinicians who are hopeful but realistic about the prospects of treatment. Care that met these simple and clearly stated priorities tended to be restricted to often limited periods of treatment by specialist "personality disorder" services: generic and primary care services were often reported as far from adequate. There is an urgent need to co-design and test strategies for improving long-term support and treatment care for people with "personality disorders" throughout the mental health care system.

## Introduction

The prevalence of "personality disorder" diagnoses is high amongst people using community and outpatient services in Europe and the USA, with estimates ranging between 40 and 92% [1]. Despite such significant levels of potential need and help-seeking, many concerns remain about the quality and accessibility of services for people given this diagnosis [2]. Stigmatising attitudes among clinical professionals in both health and mental health settings and a lack of therapeutic optimism are identified as some of the significant obstacles to the development and delivery of effective services [3–5].

A note on terminology: a contentious question in this area is regarding the value and impact of diagnosis. A substantial literature, including service user commentaries, discusses some advantages of making a diagnosis of "personality disorder" in terms of clear explanations for service users and reliable categorisation for research. Balanced against this are serious critiques of diagnoses of "personality disorder" as stigmatising and potentially misogynistic, and of the lack of progress in delivering effective care that has been associated with its use. Given the seriousness of critiques of diagnostic labels, we have chosen in this paper to use an alternative term—complex emotional needs (CEN) to describe needs often associated with a diagnosis of "personality disorder". Nonetheless, the literature that we have reviewed largely refers to "personality disorder" [6–8].

In England, effective delivery of specialist care for people with a CEN diagnosis became a priority in the early 2000s with the publication of "Personality Disorder: No Longer A Diagnosis Of Exclusion" [9] and the initiation of a set of pilot projects to establish best models of community care [9]. Fast forward to 2017 and findings from a national survey suggested that there had been up to a fivefold increase in the number of organisations providing dedicated services for people with this diagnosis [2]. However, many service users with CEN continued to face difficulties accessing good quality treatment in the community, either from specialist or generic mental health services, and the availability and nature of services remained highly variable [2]. This has resulted in a renewed policy focus on transforming care for CEN in England, and in congruent recommendations from professional bodies [10]. Policy and guideline development aimed at achieving effective and acceptable care is now identified as a priority in England. Similar needs have been identified elsewhere, including in Australia and much of Europe [11, 12].

The design and delivery of care pathways and treatments to address successfully the needs of for people with CEN needs to be informed by service users and their families and friends, as

well as by scientific evidence and professional expertise. A 2008 Delphi survey on community-based services for people with CEN found only 39% agreement amongst academic experts, service providers and services users with regards to the organisation and delivery of care [13], highlights the complexity of designing services that are satisfactory to all stakeholders and the importance of including service user perspectives in service development [14].

Involvement of service users, who are experts by experience, in service co-design is an increasingly important component of public policy and mental health system development [15, 16]. Evidence from qualitative research into service user and carer views is potentially a useful adjunct to this, helping to bring a broad range of views and experiences from different contexts to service development. Two recent systematic reviews have presented relevant summaries of such evidence. In 2017, Katsakou and Pistrang reviewed evidence on the recovery experiences of people receiving treatment for CEN, reporting service user perspectives on helpful and unhelpful service characteristics [14]. Characteristics of services facilitating helpful change included a focus on providing a safe and containing environment, and on establishing a trusting relationship between service users and clinicians. Unhelpful characteristics included placing too much emphasis on achieving change and failing to achieve collaborative therapeutic relationships. In 2019, Lamont and Dickens published a broad systematic review and meta-synthesis of service user, carer, and family experiences of all types of mental health care received by people with a diagnosis of "Borderline Personality Disorder" [17]. Overall, they found that people had clear expectations about the professional support they should receive from services, including professionalism, clinical knowledge, respect, compassion, effective interventions, and positive and non-stigmatising attitudes from professionals. However, these expectations were frequently unmet. Instead, people felt that services were frequently confusing and encounters with professionals often problematic.

The current review, conducted primarily to inform development of NHS England specialist pathways, complements and extends the above with a specific focus on community, as opposed to crisis and inpatient services, aiming to synthesise literature on service user views relevant to understanding what constitutes good care in such settings.

## Materials and methods

### Aims

To systematically review and synthesise qualitative literature on the experiences of service users with complex emotional needs (CEN) of community mental health care, and their views about what constitutes good quality care.

### Search strategy and selection criteria

The CRD handbook guidance (https://www.york.ac.uk/media/crd/Systematic_Reviews.pdf) and the PRISMA reporting guidelines were followed [18, 19]. The protocol was prospectively registered on PROSPERO (CRD42019142728). The present review was part of the NIHR Mental Health Policy Research Unit's work programme on CEN, which included four systematic reviews (alongside the current review are reviews of qualitative studies of clinician experiences, quantitative studies of service outcomes, and economic evidence of cost-effectiveness). The protocol for the wider programme of work was also registered on PROSPERO (CRD42019131834). A single search strategy was used for the whole programme, and articles relevant to each review retrieved from the resulting pool of papers. The protocol was developed by the review team in collaboration with a working group of lived-experience researchers and subject experts.

Searches of MEDLINE (January 2003—December 2019), Embase (January 2003—December 2019), HMIC (January 2003 –December 2019), Social Policy and Practice (January 2003 –

December 2019), CINAHL (January 2003—December 2019) and ASSIA (January 2003—January 2019) were conducted. The search strategy was supplemented with forward and backward citation searches of included articles. An additional search of EMBASE and MEDLINE (January 2003-November 2019) was performed to identify related systematic reviews, and the reference lists of relevant reviews were checked. Grey literature was identified through web searches and through searches of the above bibliographic databases. The full search strategy was peer reviewed using the PRESS checklist prior to searching and is available [20], including a search narrative [21], in the (S1 Text in S1 File).

Citations retrieved during searches were collated in Endnote and duplicates were removed [22]. As a single search strategy was used for a wider programme of work, initially titles and abstracts were independently double screened for all reviews simultaneously. Full text screening was then performed for citations that were potentially eligible for this review by AE and LSR. All papers thought to meet inclusion criteria and 20% of ineligible papers were double screened. In cases of disagreement or uncertainty, consensus was achieved through discussion with senior reviewers (SJ and SO).

We included primary research studies published since 2003, when "Personality Disorder: No Longer A Diagnosis Of Exclusion" was published [6], as papers that are older than this may be less relevant to current needs. No limits were placed on the language or location of publications. Eligible studies were those that:

a. Included recognised qualitative data collection and analysis methods. Written data from questionnaires were included if a recognised qualitative analysis method was used such as thematic analysis. Mixed-method studies were included if the qualitative data were reported separately to the quantitative data.

b. Reported data from adults (aged 16 or over) with a "personality disorder" diagnosis. We also considered for inclusion papers focusing on care provided for complex emotional needs described as repeated self-harm, suicide attempts, complex trauma or complex PTSD, and emotional dysregulation or instability: we made this decision as we were aware that otherwise some papers may be missed because authors and/or participants are reluctant to use the term "personality disorder" for the reasons outlined in the introduction. When the above search terms resulted in retrieval of potentially relevant papers, a group including a senior psychiatrist reviewed study context, inclusion criteria, and sample description to assess whether the majority of the sample fitted the clinical picture associated with "personality disorder".

c. for such studies we considered in each case whether the sample appeared to consist mainly of people with long-term difficulties similar to those that may result in a "personality disorder" diagnosis.

d. Data extracted for this review related to care provided by community based mental health services, including primary mental health care services, generic community mental health teams, and specialist services for people with complex emotional needs. Data related to care from residential, forensic, crisis services, or from specialist services for different conditions, such as substance misuse clinics, were excluded.

A more in-depth description of the eligibility criteria is contained in the (S2 Text in S1 File).

## Quality assessment and analysis

Data on the key characteristics of eligible studies were independently extracted by two reviewers (AE and LSR) using an Excel-based form. Quality assessment of included papers was

performed using the Critical Appraisal Skills Programme (CASP) Qualitative Checklist by two researchers [23]. Any discrepancies were resolved through discussion. Study quality was not used in decisions about eligibility but is reported and incorporated into the meta-synthesis.

Data were analysed using thematic synthesis [24]. In the first stage, preliminary codes were developed, focusing on themes relevant to understanding service user views about what constitutes good care. To do this, all relevant material was coded as examples of poor practice or care also provide information, implicitly, of good care. Two researchers inductively line-by-line coded 10 articles each and a third researcher independently second coded 50% of these. Codes were then compared and discussed between researchers until an initial set of codes was developed. The remaining articles were then divided between the three researchers for coding. New codes were added as necessary. In stage 2, an initial thematic framework was developed. Through discussion, a team of five researchers explored similarities and differences between the codes, and individual codes were split or merged as necessary. This team comprised academic and lived experience researchers. Codes were then grouped and arranged into a hierarchy to create a framework of descriptive themes. This was an iterative process involving meetings and discussion by email, and checking the framework against the original data. In the third stage, analytic themes were generated, and the framework was finalised by the research team. Towards the end of this process, the analysis was discussed with the project working group to guide interpretation of the final results. The working group was made up of 29 members with academic, lived experience, and clinical backgrounds.

## Results

We identified 47 eligible papers (Fig 1), which reported data from 44 studies and included 1,531 service users. 28 papers reported data from people diagnosed with "Borderline Personality Disorder", 12 from a sample of people with mixed "personality disorder" diagnoses, four from mixed samples of people either with a diagnosis of a "personality disorder" or who self-identified with the diagnosis, two from service users of a specialist service for "personality disorders" but did not otherwise report diagnostic or symptom information, and one from people with a history of repeated self-harm. 19 papers reported data on service user experiences of generic mental health services or of mental health care overall, 15 of specialist CEN services, 10 of specific psychotherapies, and three of independent or third sector services. Settings were the United Kingdom (n = 28), elsewhere in Europe (n = 8), Australia (n = 5), the United States (n = 3), and the rest of the world (n = 3). A summary of the included studies can be found in Table 1.

Overall, the included papers reported adequate detail for many of the topics covered by the CASP quality appraisal tool (Table 2). All included a clear statement of the aims of the research, an appropriate qualitative methodology, an appropriate recruitment strategy, and a clear statement of findings. However, a minority of papers did not include any or not enough information to determine whether the research design (n = 5), data collection method (n = 1), or the analysis method (n = 2) were appropriate to the study aims. Finally, a substantial number of papers (n = 14) did not adequately take ethical issues into consideration, while most (n = 27) failed to explore the relationship between researchers and participants adequately.

The included studies covered a range of contexts, sample populations, and approaches to data collection and analysis. The main over-arching themes from this literature are described below. But given its complexities, a fuller report is contained in the (S3 Text in S1 File). Clear differences were identified between different types of setting and levels of care (e.g. specialist versus generic and primary care), as well as between clinician groups (e.g. General Practitioners versus clinicians in specialist care). However, we were unable to identify obvious between-

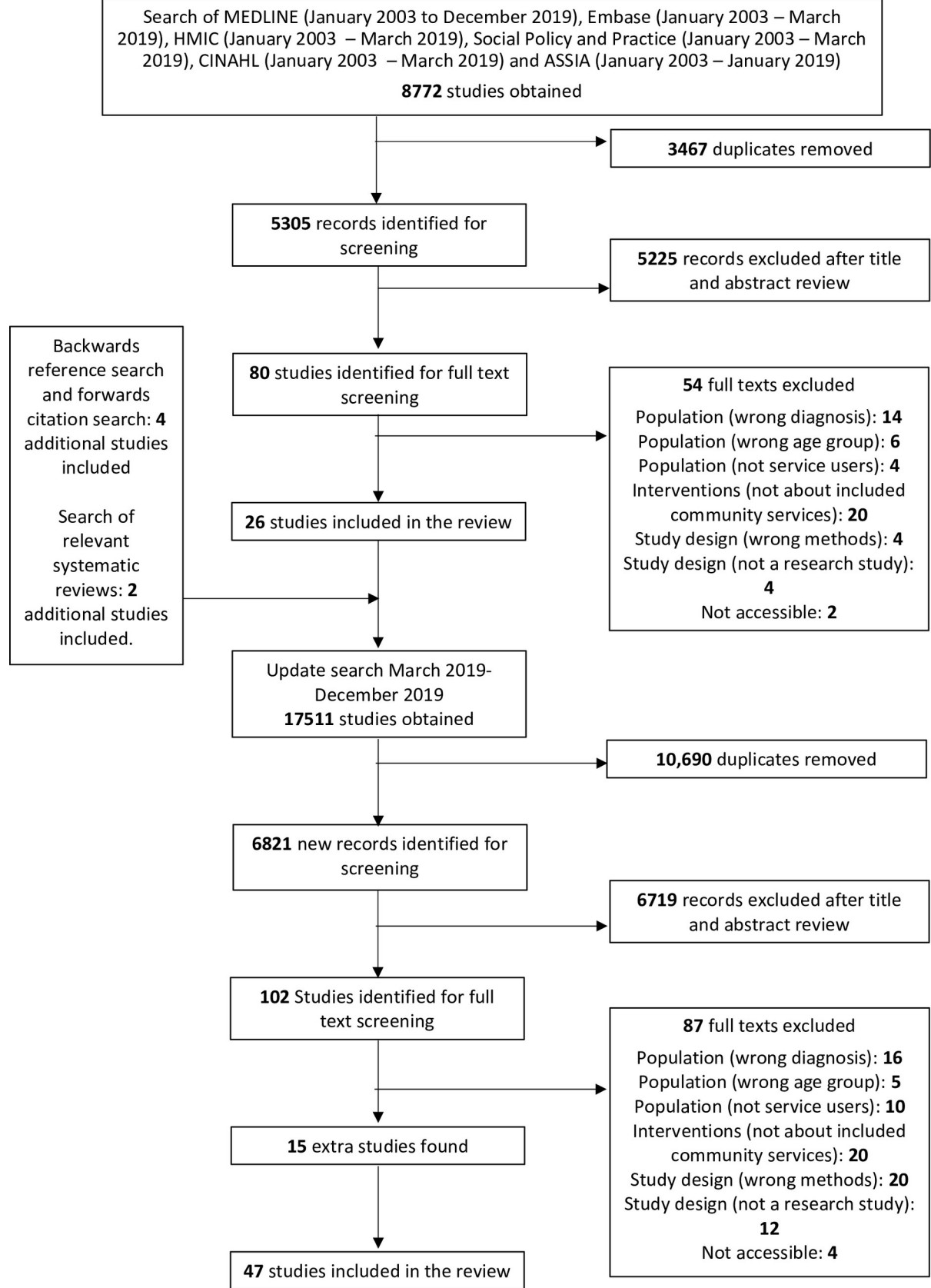

**Fig 1. PRISMA diagram.**

country differences; this is likely to be at least in part because most included studies were conducted in the UK, with other countries represented by relatively few papers. Quotes representative of themes are presented in Table 3.

## a) The need for a long-term perspective on treatment journeys

Many studies emphasised the need for a long-term perspective on treatment, supporting gradual improvement over several or many years. Service users tended to report that, although treatment benefits could accrue over time, difficulties in managing emotions and in relationships and daily living also fluctuated, so that progress made was rarely linear.

**Gradual change in awareness of and response to emotions.** Service users reported gradual improvements in emotional regulation as they gained awareness of emotions underpinning behaviours such as self-harm, resulting in a growing sense of control over these behaviours. Intensely experienced emotions included anger, sadness, anxiety, fear, hopelessness and emptiness. Recognising these emotions and being able to respond to them in different ways was described as an important component of recovery. Many pathways to change were described, including psychotherapy, art therapy and life story work. Some service users described a "light bulb" [49] moment when identifying triggers for self-harm following repeated behavioural analysis, helping the adoption of different coping strategies.

Service users described the skills learnt in treatment slowly becoming 'second nature' through small incremental steps. However, there were also many accounts of setbacks, with overwhelming emotions a barrier to effective use of newly learned coping strategies. There were accounts of a process of "personalisation", with individuals identifying the strategies that work best for them to control emotions and function effectively in day-to-day interactions.

**Gradual improvements in relationships.** A further benefit described in several studies as accruing over time was in relationships with others, as treatment enhanced realistic understanding of others' behaviour and feelings. Treatment could support service users to make more balanced assessments of others' behaviour, and to be more mindful of how their own behaviour may be experienced by others. Good therapeutic alliances were described as promoting positive changes in relationships, as were relationships with peers in services, allowing communication skills to be practiced and refined. Accounts were given of relationships with friends and family gradually becoming stronger alongside good quality care.

**Recovery.** Service users in many studies reported mixed feelings about the idea of recovery as a primary goal for services and the attendant implication of incremental progress over time, whether towards clinical or personal goals. A widely reported view was that a realistic recovery goal wasn't the absence of difficulties, but rather an improved ability to cope with them, or a reduction in their negative impacts on their lives. A pattern of periods of improvement interspersed with setbacks was often described. Clinicians were often perceived as having expectations of relatively swift recovery that were at odds with service use experiences regarding the pace and consistency of change.

## b) The need for individualised and holistic care

Service users in many studies emphasised the importance of individualised care and of availability of different types of help, rather than clinicians adopting a one-size-fits-all approach. This was reported to be a problem when clinicians focused too much on diagnosis or relied too heavily on delivering recommended and highly standardised therapies, such as DBT. The

**Table 1. Characteristics of included studies.**

| First Author | Population characteristics (age, gender, country) | Treatment and diagnosis | Data collection and analysis | Main themes |
|---|---|---|---|---|
| Barnicot et al., 2015 [25] | N = 40<br>Age: Mean = 33 (SD = 10.2)<br>Sex: 85% Female<br>United Kingdom | Dialectical behavioural therapy (DBT), 12 week individual sessions<br>Personality disorders | Semi-structured interview<br>Thematic analysis | a) Difficulty learning the skills (anxiety, too much information/jargon)<br>b) Difficulties putting the skills into practice (loss of control, negative thoughts)<br>c) Personal journey to a new life (overcoming initial difficulties, committing to change, personalising skill, skills becoming habitual)<br>d) Environment that supports change (others in the group, therapist, friends and family) |
| Barr, Hodge, & Kirkcaldy, 2008 [26] | N = 23<br>Age: 17–66<br>Sex: 17/20* (85%) female<br>United Kingdom | a) Therapeutic Community Day Services<br>b) Personality Disorders | Semi-structured interviews<br>Thematic analysis | a) The lives of service users: problem issues and areas<br>b) Experiences of the therapeutic community day services |
| Bradbury, 2018 [27] | N = 8<br>Age: 21–54<br>Sex: 8 (100%) Female<br>United Kingdom | Community Mental Health Team<br>Borderline Personality Disorder | Semi-structured interview<br>Interpretative Phenomenological Analysis | a) Trust<br>b) Qualities of the care coordinator<br>c) The complexity of the relationship<br>d) Developing a safe base |
| Carrotte, Hartup, & Blanchard, 2019 [28] | N = 9<br>Age: not reported<br>Sex: 6 (75%) Female<br>Australia | Treatment or support service for personality disorder<br>Borderline personality disorder | Semi-structured interview and focus groups<br>Thematic (framework) analysis | a) Identity and discovery<br>b) (Mis)communication<br>c) Complexities of care<br>d) Finding what works (for me)<br>e) An uncertain future |
| Castillo, Ramon, & Morant. 2013 [29] | N = 66<br>Age: 18–65<br>Sex: 47 (71%) Female<br>United Kingdom | Combined Day and Residential Respite Service<br>Personality Disorders | Semi-structured interview and focus group<br>Thematic analysis | a) Mapping the process of recovery<br>b) A sense of safety and building trust<br>c) Feeling cared for and creating a culture of warmth<br>d) A sense of belonging and community<br>e) Learning the boundaries–love is not enough<br>f) Containing experiences and develop skills<br>g) Hopes, dreams and goals and their relationship to recovery<br>h) Achievements, identity and roles<br>i) Transitional recovery and how to maintain healthy attachment |
| Chatfield, 2013 [30] | N = 6<br>Age: Not reported<br>Sex: Not reported<br>United Kingdom | Service specialising in psychodynamic interventions<br>Personality disorders | Semi-structured interviews and focus group<br>Constructivist grounded theory | a) Hope<br>b) External demands<br>c) Waiting list<br>d) Expectations of therapy<br>e) Knowledge of therapy<br>f) Experience of therapy<br>g) Information vacuum |
| Ciclitira et al., 2017 [31] | N = 59<br>Age: 23–67<br>Sex: 59 (100%) Female<br>United Kingdom | Women's Community Health Centre<br>Symptoms including self-harm, suicidal ideation, complex trauma) | Semi-structured interview<br>Thematic analysis | a) Violence and loss in the context of female oppression<br>b) A sanctuary for women<br>c) Non-medicalised long-term counselling in a safe setting<br>d) Benefits of the long view |
| Clarke, 2017 [32]<br>Same sample as Clarke 2018 | N = 21<br>Age: Not reported<br>Sex: Not reported.<br>United Kingdom | Day Therapeutic Community<br>Personality Disorders | Narrative interviews<br>Thematic analysis | a) Empowerment through inclusion<br>b) Power through exclusion |

*(Continued)*

**Table 1.** (Continued)

| First Author | Population characteristics (age, gender, country) | Treatment and diagnosis | Data collection and analysis | Main themes |
|---|---|---|---|---|
| Clarke & Waring 2018 [33] Same sample as Clarke 2017 | N = 21 Age: Not reported Sex: Not reported United Kingdom | NHS Day Community Personality Disorders | Narrative interviews Interpretative Data Analysis | a) Inclusivity within rituals: solidarity through negative transient emotions b) Transforming negative transient emotions in to high EE c) Exclusivity within rituals: negative transient emotions reinforcing low EE |
| Crawford et al., 2007 [34] | N = 133 Age: 18 to 69 (Median = 37.2 years) Sex: 70% Female United Kingdom | 11 pilot services for 'personality disorder' across England Diagnosis not reported | Semi-structured interviews and focus groups Thematic (framework) analysis | a) Desperation and hope b) Information c) Assessment d) Diagnosis e) Early impressions |
| Cunningham, Wolbert, & Lillie, 2004 [35] | N = 14 Age: 23–61 Sex: Not reported United States | Assertive Community Treatment Borderline Personality Disorder | Semi-structured interview Not specified. | a) General reflections b) Assessment of Program Component c) Effect of DBT on Day-to-Day Life |
| Donald et al., 2017 [36] | N = 17 Age: 19–59 Sex: 15 (88%) Female Australia | Specialist Outpatient Service Borderline Personality Disorder | Semi-structured interviews Thematic analysis | a) Support from others b) Accepting the need for change c) Working on trauma without blaming oneself d) Curiosity about oneself e) Reflecting on one's behaviour |
| Falconer et al., 2017 [37] | N = 15 Age: 20–43 (Mean = 31.2) Sex: 12 (80%) Female United Kingdom | Personality Disorder Service Borderline personality disorder | Semi-structured interviews Thematic analysis | a) Visualisation helps me to express and understand myself b) Visual narrative helps me to keep track and participate c) Avatars help me take and understand another's perspective d) Allowing me to see the big picture e) Giving me distance to think clearly f) Group therapy is best, but one-to-one sessions have value too |
| Fallon, 2003 [38] | N = 7 Age: 25–45 Sex: 4 (57%) Female United Kingdom | Mental health trust Borderline Personality Disorder | Unstructured interviews Grounded theory | a) Living with BPD b) Service response c) Relationships d) Travelling through the system |
| Flynn et al., 2019 [39] | N = 131 Age: Not reported Sex: Not reported United Kingdom | Internet survey Emotionally unstable personality disorder | Internet survey (service users) and focus groups (staff) Thematic analysis | a) Being diagnosed with personality disorder b) Receiving consistent and compassionate care c) Understanding recovery in personality disorder d) Access to services e) Access to effective therapies f) Staff training and support |
| Folmo, 2019 [40] | Not reported Norway | Mentalisation based therapy Borderline personality disorder | Transcripts of therapy sessions Interpretative phenomenological analysis | a) Losing authority and losing battles b) Protecting the patient from therapy c) Leaning on the alliance in the battle of the comfort zone d) Using empathetic focus to carefully battle affect avoidance |

(*Continued*)

**Table 1.** (Continued)

| First Author | Population characteristics (age, gender, country) | Treatment and diagnosis | Data collection and analysis | Main themes |
|---|---|---|---|---|
| Gillard, Turner, & Neffgen, 2015 [41] | N = 6<br>Age: 26–65<br>Sex: 3 (50%) Female<br>United Kingdom | Specialist services with peer support groups<br>Personality disorders or self-identified as having PD-related symptoms/needs | Semi-structured interview<br>Thematic analysis | a) The internal world<br>b) The external world<br>c) Diagnosis<br>d) Recovering or discovering the self—reconciling the internal and external worlds<br>e) Recovery and discovery—doing things differently<br>f) Recovery and discovery—feeling and thinking differently. |
| Gillard et al., 2015 [42] | N = 38<br>Age: Mean = 36.3<br>Sex: 28 (74%) Female (n = 28)<br>United Kingdom | Community based support group<br>Personality disorders or self-identified as having PD-related symptoms/needs | Semi-structured interview<br>Thematic and matrix analysis | a) Access and self-referral<br>b) Peer support groups and Coping Process Theory<br>c) Service users as staff<br>d) Community-based support |
| Goldstein, 2015 [43] | N = 7<br>Age: 28–45<br>Sex: 7 (100%) Female<br>United States | Community Service Centres and Clinics<br>Personality disorders or self-identified as having PD-related symptoms/needs | Semi-structured interview<br>Not specified | a) Background Information and Presentation<br>b) Synthesis of Object Relations Material From the Interview Portion<br>c) CCRT-RAP Interview Results<br>d) My Interpersonal Responses<br>e) Interpersonal Patterns Enacted in Therapy Relationships |
| Haeyan, Kleijberg, & Hinz 2018 [44] | N = 8<br>Age: 22–50<br>(Mean = 36.75)<br>Sex: 6 (75%) Female<br>Netherlands | Outpatient treatment unit for personality disorder<br>Personality disorders (borderline, avoidance, obsessive-compulsive, narcissistic) | Semi-structured interviews<br>Thematic analysis | a) Experiences with the art assignments<br>b) Material handling/interaction<br>c) Preferred approach in the art process and the Expressive Therapies Continuum level<br>d) Preferred approach in the art process and emotion regulation<br>e) Therapeutic value of the combination of factors |
| Helweg-Joergensen et al., 2019 [45] | N = 16<br>Age: Mean = 28.0<br>(SD = 6.2)<br>Sex: Not reported<br>Denmark | Public outpatient psychiatric care<br>Emotionally unstable personality disorder | Focus groups<br>Grounded theory | a) Barriers and facilitators<br>b) Balancing acceptance and change during inside-out innovation |
| Hodgetts, Wright, & Gough, 2007 [46] | N = 5<br>Age: 24–48<br>(Mean = 35.6)<br>Sex: 3 (60%) Female<br>United Kingdom | Dialectical Behaviour Therapy service<br>Borderline personality disorder | Semi-structured interview<br>Interpretative Phenomenological Analysis | a) Joining a DBT Programme (external and internal factors)<br>b) Experience of DBT (specific and non-specific factors)<br>c) Evaluation of DBT (change, evaluation and role of the past and future) |
| Hummelen, Wilberg, & Karterud 2007 [47] | N = 8<br>Age: 24–48<br>(Mean = 35.6)<br>Sex: 8 (100%) Female<br>Norway | Psychotherapeutic day hospitals<br>Borderline personality disorder | Semi-structured interview<br>Not specified | a) Difficult transition<br>b) Group therapy was too distressing<br>c) Outpatient group therapy was insufficient<br>d) Not able to make use of the group<br>e) Complicated relationship to the group<br>f) Negative aspects of the patient–therapist relationship<br>g) Too much external strain<br>h) Desire to escape from therapy<br>i) No interest in further long–term group therapy<br>j) Reasons not mentioned by the patients |

(*Continued*)

**Table 1.** (*Continued*)

| First Author | Population characteristics (age, gender, country) | Treatment and diagnosis | Data collection and analysis | Main themes |
|---|---|---|---|---|
| Katsakou et al. 2012 [48] Same sample as Katsakou et al. 2017 | N = 48 Age: Mean = 36.5 Sex: 39 (81%) Female United Kingdom | Specialist Services including community mental health teams and psychological therapies Personality Disorders | Semi- structured interview Thematic Analysis & Grounded Theory | a) Personal goals and/or achievements during recovery b) Balancing personal goals of recovery versus service targets c) How recovered do people feel? d) Problems with the word 'recovery' |
| Katsakou et al. 2017 [49] Same sample as Katsakou et al. 2012 | N = 48 Age: Mean = 36.5 Sex: 39 (81%) Female United Kingdom | Specialist Services including community mental health teams and psychological therapies Borderline Personality Disorder | Semi-structured interview Thematic analysis | a) Processes of recovery: Fighting ambivalence and committing to taking action. Moving from shame to self-acceptance and compassion. Moving from distrust and defensiveness to opening up to others. b) Challenges in therapy: Balancing self-exploration and finding solutions. Balancing structure and flexibility. Confronting interpersonal difficulties and practicing new ways of relating. Balancing support and independence. |
| Larivière et al., 2015 [50] | N = 12 Age: 23–63, (Mean = 37.2; SD = 13.3) Sex: 12 (100%) Female Canada | Not reported Borderline personality disorder | Picture collage and semi-structured interview Thematic analysis | a) Living with borderline personality disorder b) Dimensions of recovery (related to the person and the environment) c) Facilitators |
| Leung et al., 2019 [51] | N = 11 Age: 24–58 Sex: 9 (82%) Female China | Emergency medical ward History of self-harm | Semi-structured interview Thematic analysis | a) Service availability b) Accessibility c) Affordability d) Acceptability |
| Lohamn et al., 2017 [52] | N = 500 Age: 18 and above Sex: Not reported United States | Borderline Personality Disorder Resource Centre Borderline Personality Disorder | Written or Telephone transcripts of previously collected data through unstructured interviews Conventional Qualitative Content Analysis | a) Requested Services b) Mental Health Literacy and Marginalization c) Family and Caregiver Resources d) Insurance and Finances e) Medical and Psychiatric Comorbidity f) Crisis Services |
| Lonargain, Hodge, & Line 2017 [53] | N = 7 Age: 26–52 (Mean = 39.9) Sex: 5 (71%) Female United Kingdom | Mentalisation-based therapy (MBT) groups Borderline personality disorder | Semi-structured interview Interpretative Phenomenological Analysis | a) Experiencing group MBT as unpredictable and challenging b) Building trust: a gradual but necessary process during MBT c) Putting the pieces together: making sense of the overall MBT structure d) Seeing the world differently due to MBT: a positive shift in experience+ |
| Long, Manktelow, & Tracey 2016 [54] | N = 10 Age: 19–42 (Mean = 31) Sex: 8 (80%) Female United Kingdom | Not reported Borderline personality disorder | Semi-structured interview Grounded theory | a) Building up trust b) Seeing beyond the cutting c) Human contact d) Integrating experiences |
| McSherry et al., 2012 [55] | N = 30 Age: 32–55 Sex: Not reported Ireland | Community Adult Mental Health Borderline Personality Disorder or self-identified as having PD-related symptoms/needs | Semi-structured interview and focus group Thematic analysis | a) Evaluation of therapy b) Treatment impact |

(*Continued*)

**Table 1.** (Continued)

| First Author | Population characteristics (age, gender, country) | Treatment and diagnosis | Data collection and analysis | Main themes |
|---|---|---|---|---|
| Morant & King, 2003 [56] | N = 15<br>Age: Not specified<br>Sex: Not specified<br>United Kingdom | Outreach Service Team<br>Personality Disorders | Semi-structured interview<br>Content thematic analysis | Not clearly specified |
| Morris, Smith, & Alwin, 2014 [57] | N = 9<br>Age: 18–65<br>Sex: Not reported<br>United Kingdom | Adult Mental Health Services<br>Borderline Personality Disorder | Semi-structured interview<br>Thematic analysis | a) The diagnostic process influences how service users feel about BPD<br>b) Non-caring care<br>c) It's all about the relationship |
| Mortimer-Jones et al., 2019 [58] | N = 8<br>Age: Not reported<br>Sex: 7 (88%) Female<br>Australia | Short term residential service<br>Borderline personality disorder | Semi-structured interview<br>Inductive phenomenological analysis | a) Benefits of the programme<br>b) Enhanced client outcomes<br>c) Impact of the physical environment<br>d) Ways of enhancing the service |
| Naismith et al., 2019 [59] | N = 53<br>Age: 18–57<br>(Mean = 32;<br>SD = 11.1)<br>Sex: 44 (83%) Female<br>United Kingdom | Outpatient personality disorder service<br>Personality disorders (borderline, narcissistic, not specified) | Focus group<br>Thematic analysis | a) Experience of treatment: compassion, relaxation, difficult, negative emotions<br>b) Inhibitors: weak imagery ability, fear of compassion, lack of compassionate experiences, distressing affect/cognitions, lack of distress, psychological symptoms |
| Ng et al., 2019a [60] | N = 102<br>Age: 18–56<br>(Mean = 29.7;<br>SD = 8.84)<br>Sex: 89 (87%) Female<br>Australia | Community-based psychotherapy programme<br>Borderline personality disorder | First assessment session for treatment<br>Inductive conventional content analytic approach | a) Reducing symptoms<br>b) Improve well-being<br>c) Better interpersonal relationships<br>d) Greater sense of self |
| Ng et al., 2019b [61] | N = 14<br>Age: 18–52<br>(Mean = 33.26;<br>SD = 10.26)<br>Sex: 14 (100%) Female<br>Australia | Online survey by mental health organisations<br>Borderline personality disorder | Semi-structured interview<br>Interpretive phenomenological analysis | a) Stages of recovery (Being stuck, Diagnosis, Improving experience)<br>b) Developing greater awareness of emotions and thoughts<br>c) Strengthening sense of self<br>d) Developing greater awareness of emotions and thoughts<br>e) Processes of recovery in borderline personality disorder<br>f) Active engagement in the process of recovery<br>g) Hope<br>h) Engagement with treatment services<br>i) Engaging in meaningful activities and relationships |
| Perseius et al, 2003 [62]<br>Same sample as Perseius, 2005 | N = 10<br>Age: 22–49,<br>(Median = 27)<br>Sex: 10 (100%) Female<br>Sweden | Outpatient treatment<br>Borderline personality disorder | Individual, focused interview<br>Content analysis | a) The therapy is life-saving<br>b) The therapy provides skills to help conquer suicidal and self-harm impulses<br>c) Respect and confirmation is the foundation<br>d) The method of therapy-brings understanding and focus on the problems<br>e) Your own responsibility and the stubborn struggle with yourself<br>f) The therapy contract brings support and challenge<br>g) The group therapy—hard but necessary<br>h) The telephone coaching–important crises support<br>i) Not being understood and disrespectful attitudes<br>j) Discontinuity and betrayal<br>k) The poorly adapted tools of psychiatric care |

*(Continued)*

**Table 1.** (Continued)

| First Author | Population characteristics (age, gender, country) | Treatment and diagnosis | Data collection and analysis | Main themes |
|---|---|---|---|---|
| Perseius et al., 2005 [63] Same sample as Perseius, 2003 | N = 10 Age: 22–49, (Median = 27) Sex: 10 (100%) Female Sweden | Outpatient treatment for self-harming Borderline personality disorder | Narrative interviews, supplemented by biographical material Hermeneutic approach | a) Life on the edge b) Struggle for health and dignity c) The good and the bad act of psychiatric care in the drama of suffering |
| Rogers & Acton, 2012 [64] | N = 7 Age: 21–43 Sex: 6 (86%) Female United Kingdom | Specialist service for personality disorder Borderline personality disorder | Semi-structured interview Thematic analysis | a) Staff knowledge and attitudes b) Lack of resources c) Recovery pathway |
| Rogers & Dunne, 2013 [65] | N = 7 Age: 21–61 Sex: 5 (71%) Female United Kingdom | Specialist Personality Disorder Service Personality Disorders | Focus groups Thematic analysis | a) Having a Voice b) Progression versus Consistency c) Moving On from Services d) Understanding Personality Disorder e) Understanding Recovery f) Lack of Information g) Follow Up h) Accessing Treatment |
| Sheperd, Sanders, & Shaw, 2017 [66] | N = 17 Age: 31–60 Sex: 12 (71%) Female United Kingdom | General Community Service Personality Disorders | Semi-structured interview Thematic analysis | a) Understanding early lived experience as informing sense of self b) Developing emotional control c) Diagnosis as linking understanding and hope for change d) The role of mental health services. |
| Smith, 2013 [67] | N = 6 Age: 22–30 (Mean = 26) Sex: 6 (100%) Female United Kingdom | NHS community-based DBT programme Borderline personality disorder | Semi-structured interview Interpretive phenomenological analysis | a) Therapeutic Group Factors b) Therapist factors c) Personal change d) Challenges to be overcome e) Personalised problem solving f) Opposing expectations |
| Stalker, Ferguson, & Barclay, 2010 [68] | N = 10 Age: 27–52 Sex: 8 (80%) Female United Kingdom | Mental Health Resource Centres Personality disorder | Semi-structured interview Grounded theory | a) Understanding of personality disorder b) Perceived helpfulness of the diagnosis c) Difficulties faced by people with a personality disorder diagnosis d) Perceived causes of people's difficulties e) What helps? |
| van Veen et al., 2019, [69] | N = 13 Age: 20–60 Sex: 11 (85%) Female Netherlands | Outpatient services Personality disorders (Borderline, Obsessive compulsive, Avoidant, Dependent)) | Semi-structured interview Grounded theory | a) Goals that were mutually agreed on b) Mutually agreed-on tasks c) The interpersonal relationship between the CMHN d) and the patient |
| Veysey, 2013, [70] | N = 8 Age: 25–65 Sex: 6 (75%) Female New Zealand | Mental health awareness newsletters Borderline personality disorder | Semi-structured interview Interpretive phenomenological analysis | e) Self-harm and discriminatory experiences f) Negative messages about BPD g) Negative impact on self-image h) Stigma and complaints i) Helpful behaviour: connecting; seeing more j) Individuals have an impact k) Contrasting ideas |
| Walker, 2009, [71] | N = 4 Age: 30–54 Sex: 4 (100%) Female United Kingdom | Community centres Borderline Personality Disorder | Narrative interview Narrative thematic analysis | a) 'Self-harm'—seeing beyond the scars b) 'Being known' as a self-harmer |

* Data incomplete.

**Table 2. Quality assessment of the studies.**

| | 1. Are the results valid? | 2. Is a qualitative methodology appropriate? | 3. Was the research design appropriate to address the aims of the research? | 4. Was the recruitment strategy appropriate to the aims of the research? | 5. Was the data collected in a way that addressed the research issue? | CASP Item 6. Has the relationship between researcher and participants been adequately considered? | 7. Have ethical issues been taken into consideration? | 8. Was the data analysis sufficiently rigorous? | 9. Is there a clear statement of findings? | 10. How valuable is the research? |
|---|---|---|---|---|---|---|---|---|---|---|
| Barnicot et al., 2015 [25] | Yes | Yes | Yes | Yes | Yes | Yes | Can't tell | Yes | Yes | Valuable |
| Barr, Hodge, & Kirkcaldy, 2008 [26] | Yes | Yes | Yes | Yes | Yes | Can't tell | Can't tell | Yes | Yes | Valuable |
| Bradbury, 2018 [27] | Yes | Yes | Yes | Yes | Yes | Can't tell | Yes | Yes | Yes | Valuable |
| Carrotte, Hartup, & Blanchard, 2019 [28] | Yes | Yes | Can't tell | Yes | Yes | No | No | Yes | Yes | Valuable |
| Castillo, Ramon, & Morant. 2013 [29] | Yes | Yes | Yes | Yes | Yes | No | Can't tell | Can't tell | Yes | Valuable |
| Chatfield, 2013 [30] | Yes | Yes | Yes | Yes | Yes | Yes | Yes | Yes | Yes | Valuable |
| Ciclitira et al., 2017 [31] | Yes | Yes | Can't tell | Yes | Yes | Yes | Yes | Yes | Yes | Valuable |
| Clarke, 2017 [32] | Yes | Yes | Yes | Yes | Can't tell | No | No | No | Yes | Valuable |
| Clarke & Waring 2018 [33] | Yes | Yes | Yes | Yes | Yes | No | Can't tell | Yes | Yes | Valuable |
| Crawford et al., 2007 [34] | Yes | Yes | Yes | Yes | Yes | No | Yes | Yes | Yes | Valuable |
| Cunningham, Wolbert, & Lillie, 2004 [34] | Yes | Yes | Yes | Yes | Yes | Yes | Yes | Yes | Yes | Valuable |
| Donald et al., 2017 [36] | Yes | Yes | Yes | Yes | Yes | No | No | Yes | Yes | Valuable |
| Falconer et al., 2017 [37] | Yes | Yes | Yes | Yes | Yes | No | No | Yes | Yes | Valuable |
| Fallon, 2003 [38] | Yes | Yes | Yes | Yes | Yes | No | Yes | Yes | Yes | Valuable |
| Flynn et al., 2019 [39] | Yes | Yes | Can't tell | Yes | Yes | No | No | Can't tell | Yes | Valuable |
| Folmo, 2019 [40] | Yes | Yes | Yes | Yes | Yes | No | Yes | Yes | Yes | Valuable |
| Gillard, Turner, & Neffgen, 2015 [41] | Yes | Yes | Yes | Yes | Yes | Yes | Yes | Yes | Yes | Valuable |
| Gillard et al., 2015 [42] | Yes | Yes | Yes | Yes | Yes | No | Yes | No | Yes | Valuable |
| Goldstein, 2015 [43] | Yes | Yes | Yes | Yes | Yes | Yes | Yes | Yes | Yes | Valuable |
| Haeyan, Kleijberg, & Hinz 2018 [44] | Yes | Yes | Yes | Yes | Yes | Yes | Can't tell | Yes | Yes | Valuable |
| Helweg-Joergensen et al., 2019 [45] | Yes | Yes | Yes | Yes | Yes | No | No | Yes | Yes | Valuable |
| Hodgetts, Wright, & Gough, 2007 [46] | Yes | Yes | Yes | Yes | Yes | No | No | Yes | Yes | Valuable |
| Hummelen, Wilberg, & Karterud 2007 [47] | Yes | Yes | Can't tell | Yes | Yes | No | No | Yes | Yes | Valuable |
| Katsakou et al. 2012 [48] | Yes | Yes | Yes | Yes | Yes | No | Yes | Yes | Yes | Valuable |
| Katsakou et al. 2017 [49] | Yes | Yes | Yes | Yes | Yes | Yes | Yes | Yes | Yes | Valuable |
| Larivière et al., 2015 [50] | Yes | Yes | Yes | Yes | Yes | No | Yes | Yes | Yes | Valuable |
| Leung et al., 2019 [51] | Yes | Yes | Yes | Yes | Yes | No | No | Yes | Yes | Valuable |
| Lohamn et al., 2017 [52] | Yes | Yes | Yes | Yes | Yes | No | Yes | Yes | Yes | Valuable |
| Lonargain, Hodge, & Line 2017 [53] | Yes | Yes | Yes | Yes | Yes | Yes | Yes | Yes | Yes | Valuable |
| Long, Manktelow, & Tracey 2016 [54] | Yes | Yes | Yes | Yes | Yes | No | Yes | Yes | Yes | Valuable |
| McSherry et al., 2012 [55] | Yes | Yes | Can't tell | Yes | Yes | No | Yes | Yes | Yes | Valuable |

(*Continued*)

**Table 2.** (Continued)

| | 1. Are the results valid? | 2. Is a qualitative methodology appropriate? | 3. Was the research design appropriate to address the aims of the research? | 4. Was the recruitment strategy appropriate to the aims of the research? | 5. Was the data collected in a way that addressed the research issue? | 6. Has the relationship between researcher and participants been adequately considered? | 7. Have ethical issues been taken into consideration? | 8. Was the data analysis sufficiently rigorous? | 9. Is there a clear statement of findings? | 10. How valuable is the research? |
|---|---|---|---|---|---|---|---|---|---|---|
| | | | | | | CASP Item | | | | |
| Morant & King, 2003 [56] | Yes | Yes | Yes | Yes | Yes | No | No | No | Yes | Valuable |
| Morris, Smith, & Alwin, 2014 [57] | Yes | Yes | Yes | Yes | Yes | Can't tell | Yes | Yes | Yes | Valuable |
| Mortimer-Jones et al., 2019 [58] | Yes | Yes | Yes | Yes | Yes | No | Yes | No | Yes | Valuable |
| Naismith et al., 2019 [59] | Yes | Yes | No | Yes | Yes | No | No | Yes | Yes | Valuable |
| Ng et al., 2019a [60] | Yes | Yes | Yes | Yes | Yes | No | No | Yes | Yes | Valuable |
| Ng et al., 2019b [61] | Yes | Yes | Yes | Yes | Yes | No | No | Yes | Yes | Valuable |
| Perseius et al, 2003 [62] | Yes | Yes | Yes | Yes | Yes | Yes | Yes | Yes | Yes | Valuable |
| Perseius et al., 2005 [63] | Yes | Yes | Yes | Yes | Yes | Yes | No | Yes | Yes | Valuable |
| Rogers & Acton, 2012 [64] | Yes | Yes | Yes | Yes | Yes | Can't tell | Yes | Yes | Yes | Valuable |
| Rogers & Dunne, 2013 [65] | Yes | Yes | Yes | Yes | Yes | Can't tell | Yes | Yes | Yes | Valuable |
| Sheperd, Sanders, & Shaw, 2017 [66] | Yes | Yes | Yes | Yes | Yes | No | Yes | Yes | Yes | Valuable |
| Smith, 2013 [67] | Yes | Yes | Yes | Yes | Yes | Yes | Yes | Yes | Yes | Valuable |
| Stalker, Ferguson, & Barclay, 2010 [68] | Yes | Yes | Yes | Yes | Yes | No | Yes | Yes | Yes | Valuable |
| van Veen et al., 2019, [69] | Yes | Yes | Yes | Yes | Yes | Yes | Yes | Yes | Yes | Valuable |
| Veysey, 2013, [70] | No | Yes | Yes | Yes | Yes | Yes | Yes | Yes | Yes | Valuable |
| Walker, 2009, [71] | Yes | Yes | Yes | Yes | Yes | Yes | Yes | Yes | Yes | Valuable |

**Table 3. Table of quotes.**

| Main Theme | Sub-theme | References associated with theme/sub-theme | Quote* | Source |
|---|---|---|---|---|
| The Need For A Long Perspective On Treatment journeys | Changes Over Time | 25, 30, 44, 46, 48, 49, 54, 55, 58 | 'Recovery was experienced as a series of achievements and setbacks, as SUs moved back and forth between these two poles of each recovery process. During this movement, they usually maintained an overall sense of moving forward, despite setbacks.' | Katsakou et al., 2017, [47] |
| | | | '"I found that I was doing the same thing over and over again . . . unless you understand yourself I don't think that . . . you can recover."' | Gillard et al., 2015, [39] |
| | Gradual Change In Awareness And Response To Emotions | 25, 26, 29, 30, 31, 34, 35, 36, 37, 38, 40, 41, 43, 44, 46, 47, 48, 49, 50, 53, 55, 58, 60, 61, 62, 66, 67, 70, 71 | '"I think in terms of, like, recovery, in terms of being able to have a degree of self-control and being able to think ahead about the consequences of things so that rather than having a big blow up." ' | Shepherd et al., 2017, [64] |
| | Gradual Improvements In Relationships | 26, 31, 34, 35, 37, 48, 53, 55 | '"I've got a better understanding of myself, and of other people. . . I value my emotional intelligence. . . I kind of developed it. And that's all developed in my children as well and they've got much better."' | Ciclitira et al., 2017, [29] |
| | Recovery | 25, 28, 29, 36, 41, 46, 48, 49, 50, 54, 56, 61, 62, 64, 65 | '"Yesterday was relatively ok, today is ok so far. But before, consistently, I had a period where I couldn't actually leave the house and I was very dissatisfied and self-hating. . . So it's difficult to actually trust the times when I am feeling alright."' | Katsakou et al. 2012, [46] |
| The Need For Individualised and Holistic Care | | 44, 46, 48, 49, 62, 67, 69 | 'Some participants thought that there was a clash between their personal aspirations and the focus of treatment. They felt that therapy did not address all problems they were struggling with. Some treatments were experienced as focusing almost exclusively on specific topics, i.e. self-harming or relationships (often as they were enacted in the group setting), leaving service users frustrated when they could not address other issues that were either equally or more important to them. "DBT helped, but it didn't answer all of my questions. It didn't help me to work things through myself, it didn't help me to achieve my goals really. . . I was trying to get over my divorce and also my relationship with my mum and men, and I was trying to work through it but it was all about other things, it was about self-harming, it was about mindfulness. . ."' | Katsakou et al. 2012, [46] |
| | Need For Helpful Approaches To Care | 26, 30, 34, 49, 50, 58, 61, 69 | 'A theme highlighted by a number of the service users was the positive focus of a service: the fact that it seemed to be helping them to move forwards, and that staff believed in their individual capacity for change and improvement. This was significant for the many people who had negative experiences of life as well as of mainstream services. | Crawford et al. 2007, [32] |
| | | | "This is the only service that is concentrating on getting me better, everything else seems to be just keeping me in the same place, everything else is about keeping me stable and keeping me, um, so I don't tip back over the edge. Here they're willing to push me over the edge if it involves me making steps forward."' | |
| | Medications | 34, 64, 65 | '"I just think when you first come into service that they experiment on you . . . over the course of years they've experimented with lots of different drugs. I've felt like they didn't understand, and they just like piled me with any sort of medication."' | Rogers & Acton, 2012, [62] |

*(Continued)*

**Table 3.** (*Continued*)

| Main Theme | Sub-theme | References associated with theme/sub-theme | Quote* | Source |
|---|---|---|---|---|
| Large Variations In Service Access And Quality | Access To Services | 26, 28, 30, 34, 38, 39, 46, 47, 51, 56, 57, 61, 64, 65 | "'I was struggling. . . I guess it's all a learning journey, but it would be helpful if, for me, if I had more access to stuff off the bat than having to search for it myself and figure it out myself.'" | Carrotte et al., 2019, [26] |
| | | | '[E]ven in a London-based service, there were concerns about access for people in one borough because the service was based in the other of the two boroughs it served. With the frequent mergers of NHS Trusts, catchment areas are constantly becoming larger and making access more of an issue.' | Crawford et al. 2007, [32] |
| | Quality Of Services | , 28, 31, 34, 38, 51, 57, 64, 65, 68, 71 | 'The first experience for all the participants was moving into the mental health system via referral from their general practitioner (GP). Here they met various mental health professionals, yet the explanations given were highly variable. Despite their distress and confusion some received no explanations concerning the roles of the individuals they were seeing or of the function of their contact with them.' | Fallon 2003, [36] |
| | The Role Of Specialist Services | 29, 31, 36, 39, 41, 48, 49, 50, 52, 57, 64, 65 | 'Service users generally felt they received better help and support from a specialist service for personality disorder than from community mental health teams. . . | Rogers & Acton, 2012, [62] |
| | | | The specialist service [used by participants] was also beneficial for promoting the use of alternative forms of treatment, i.e. talking therapies. The specialist service also placed an emphasis on evidence based practice, offering a treatment that is widely acknowledged to be helpful for those with the BPD diagnosis: "My DBT [Dialectical Behaviour Therapy] that I'm doing now–I've done DBT a bit on the past–but I find it more beneficial than medication for instance."' | |
| | Continuity Of Care | 27, 28, 34, 38, 39, 46, 51, 56, 65 | "'You're discharged from that service, then you're left high and dry.'" | Rogers & Dunne, 2013 [63] |
| The Centrality Of Therapeutic Relationships | | 26, 27, 28, 29, 30, 31, 34, 35, 38, 39, 40, 43, 44, 46, 47, 54, 57, 58, 59, 61, 62, 64, 65, 66, 67, 68, 69, 70, 71 | "'I felt like I didn't want to talk to them you know, if they didn't understand me they are never going to come up with something different, they are not going to turn my life around.'" | Bradbury, 2016, [25] |
| | | | "'. . ..you don't know you are unwell and the only person who is connecting with you is my care coordinator. Because she knows me inside and out all this time and although you see different psychiatrists- they do get to know you- but she has been the rock all the way and she's been the same person all the way along.'" | Bradbury, 2016, [25] |
| | Relationship Dynamics And Involvement | 27, 28, 32, 35, 38, 40, 43, 49, 55, 69 | 'Overall, participants expressed preference for therapy that was not too soft and not too hard, so to speak. They seemed to desire the allowance of moderate movement and collaborative reworking of the therapy structure and relationship but, ultimately, they did not want to call all the shots or to feel they were stronger than their therapists.' | Goldstein 2015, [41] |

(*Continued*)

**Table 3.** (Continued)

| Main Theme | Sub-theme | References associated with theme/sub-theme | Quote* | Source |
|---|---|---|---|---|
| | Family And Friends | 25, 26, 30, 34, 35, 41, 43, 48, 50, 52, 66, 67, 68, | "'I can talk to my family more effectively about what's going on in my life, whereas before I was afraid to tell them what was happening...Most of it is communication. A lack of communication gives dark thoughts.'" | Cunningham, Wolbert, & Lillie 2004, [33] |
| | Peer Support | 26, 29, 32, 33, 34, 36, 47, 49, 50, 59 | "'You realised that you weren't the only one feeling like that, there were other people in the world that felt the way that you did and being able to talk to them and hear their experiences of how they were dealing with it was helpful.'" | Crawford et al., 2007, [32] |
| | Group Treatment | 25, 26, 27, 29, 30, 32, 33, 34, 35, 36, 37, 40, 41, 42, 43, 44, 46, 47, 48, 49, 50, 53, 54, 55, 56, 57, 58, 59, 60, 61, 62, 67, | "'Ruth' reported that during her first group session she struggled with 'hard hitting' topics such as suicide and 'Sarah' recalled finding it 'difficult and scary' when she thought another group member was criticising her.' | Lonargain et al., 2017, [51] |
| Impacts Of 'Personality Disorder' Diagnosis | | 25, 26, 28, 34, 35, 39, 40, 41, 43, 46, 49, 50, 51, 52, 55, 57, 61, 66, 67, 68, 70 | "'I feel like once you get a diagnosis of BPD they sort of act like you are kind of beyond their...bother. Like they don't especially want to do anything because you are not going to be easy.'" | Bradbury, 2016, [25] |

* Quotation marks indicate a service user quote published in the paper.

importance of taking a holistic view of needs, and of focusing on personal goals and aspirations was recurrently reported. In some cases, service users reportedly felt that therapies which were the main treatments offered did not address past traumas or the problems they were struggling with in their daily lives, and instead focused almost exclusively on specific topics such as self-harm or relationships. This could leave service users frustrated that they could not address other issues that were equally or more important to them, or that help in managing the social difficulties and the challenges of everyday life was unavailable. It was also important to service users that therapy helped them adapt skills learnt in treatment to their own personal situations.

**Need for helpful approaches to care.** Another recurrently reported facet of good quality care was that it should be informed by an acknowledgement that service users often face very daunting psychological impediments to engaging with treatment. Overwhelming emotions were frequently identified as a barrier to using strategies learnt in treatment, and there were accounts of such emotions being triggered by therapy sessions. This could discourage further attendance at therapy sessions, even when service users felt it important to explore and process adverse experiences. Therapies that were primarily focused on self-harm were viewed ambiva-lently by some; self-harm was often seen as a coping strategy for dealing with unbearable emo-tional pain and distress, and could thus sometimes be experienced as life-saving rather than life-threatening. Positive or helpful approaches that were aimed at developing alternative cop-ing strategies rather than eliminating self-harm were often preferred. There were reports that having boundaries and consequences for self-harm can be helpful, but it was emphasised that such restrictions should be within a context of compassion and understanding, along with continued access to warmth and comfort from clinicians.

**Medications.** Whilst both psychological and social interventions were valued, papers tended to describe more ambivalent views about medication. There were accounts of service users feeling they were used as "guinea pigs" [64] and trialled on numerous medications because staff did not know how to treat them. This was reported in both specialist and generic

settings. Some reported being told to take new medications without any information about the rationale for this. Some papers reported service users' views medication was over-emphasised in treating CEN rather than offering psychotherapy or other treatments. Specialist services, however, were described in many papers as approaching things differently, with choices offered regarding medication use, rather than it being presented as the mainstay of treatment. Involvement in treatment decisions allowed service users some power to decide on their own recovery pathway, which varied between individuals.

### c) Large variations in accessibility and quality of mental health services

Many of the included papers focused on specialist "personality disorder" services, and many positive experiences were described of these. Services were recurrently reported to be most helpful when they were accessible and easy to understand, when staff were knowledgeable and warm, where service users were involved in their own care, such as their Care Plan Approach meetings, and where they had good access to high quality services that could offer treatment options well suited to their needs. However, there were also many accounts of complicated journeys through services and of large variations in access to and quality of care, with accounts of good care from generic mental health services being much less common.

**Access to services.**   Consistent and easy access to high quality care was highly valued but rarely reported in the included studies. Gaps in treatment pathways and exclusion from a variety of mental health services on grounds of "personality disorder" diagnosis were prominent in many of the papers that discussed the mental healthcare system beyond specialist "personality disorder" services. For many, mental health services were confusing and difficult to navigate. There were accounts of service users having to learn independently what services and treatments were available, while advocating for themselves and others as they navigated the system. Meanwhile, other service users said that they were not aware of the types of services available to them as staff had failed to signpost them.

A few papers took an overview of service provision and described large variations between areas and resource limitations. Identified barriers to access included difficulties in reaching services, particularly in rural areas or where specialist services covered large areas, poor physical facilities, high costs of specialist treatment where available, rigid inclusion/exclusion criteria, and treatment delivered mainly available through private healthcare. Temporal aspects of treatment were also important. Service users often found that starting at a new service was challenging. They experienced long wait times, found that the entry assessments were emotionally demanding, and did not understand what the service would provide. Thus, for many, the long treatment journey appeared to involve periods of reasonably good care interspersed with other periods of lacking access to any services or confusion about which pathways and services are available to them.

**Quality of services.**   There were accounts of service users receiving no explanations at all of the roles of the individuals they were seeing or of the purpose of their contact with them. Care was described that consisted of a series of rushed outpatient appointments, with service users feeling entirely excluded from important aspects of decision-making about their care. A lack of knowledgeable, engaged staff resulted in some service users feeling let down and rejected by services, especially if they did not respond to typical treatment strategies. Consequently, some service users reported looking for alternative sources of support, such as online resources, which could at times cause more harm than good, or result in greater use of problematic coping strategies.

**The role of specialist services.**   Across the papers, quality of treatment, staff attitudes and service user involvement and choice tended to be viewed as substantially better in specialist

'personality disorder' than in generic mental health services. However, specialist services were also less accessible. Furthermore, service users reported they often lacked a clear explanation of what specialist services would offer them and their input was frequently time-limited, whether because of limits imposed on the number of treatment sessions or time in the service, or through termination of treatment because 'rules' had not been adhered to: this is not in keeping with the long recovery journey discussed above.

**Continuity of care.** Lack of continuity of care was a key issue in many papers, particularly after discharge from specialist services or when key staff stopped working with a service user. Service users described needing support to maintain progress, but that relatively little help was available after the end of intensive periods of therapy. The endings of treatments could be particularly difficult for service users who were highly aware of the time limits of their service and often felt that they were too short, both in terms of length and quantity of therapy sessions. Discharge from a service could feel abrupt and result in a sudden drastic reduction in support available.

## d) The centrality of therapeutic relationships

Good client-clinician relationships were described in many sources as being at the centre of good care. Positive qualities for clinicians included being warm, trustworthy, honest, open, accepting, non-judgemental, and interested in their job and in the service user as a person. It was important to service users that they felt supported, valued, understood, listened to, and cared about. Where this was absent, service users felt that they could not be honest with their clinician, that the clinician would be unable to help them, or that their treatment would be poorly tailored to their needs. Positive qualities were described in many papers as being more frequent among clinicians working in specialist services, potentially as a result of good training and understanding of CEN.

Problematic qualities for clinicians included being poorly informed, misinformed, or perpetuating stigmatising attitudes and therapeutic nihilism (an inappropriately pessimistic view regarding the potential benefits of treatment) [72] about CEN. Some sources identified underlying problems as lack of training in working with people with CEN, poor empathy, and understanding, or a perception of people with CEN as "difficult". There were accounts of clinicians who seemed uninterested in people with CEN, who rushed through their appointments, or were dismissive, unsympathetic, or insensitive. Other negative characteristics included being overly strict, authoritarian, critical, superior, cold, or aloof. Sources often identified such experiences as most frequent in primary care, psychiatric outpatient settings or generic secondary mental health teams. When severe, service users described these experiences as traumatising. Further difficulties reported in generic settings included a lack of consistent relationships with the same professionals and a sense that clinicians had no clear therapeutic plans.

**Relationship dynamics and involvement.** Encouragement to set and work towards goals was described as important. There were accounts of service users wanting to be challenged by their clinician and pushed to progress in treatment, for example to stop self-harming. But it was also important that clinicians understood the capabilities and limits of the service user as well as the severity of their distress: and did not pushing too hard, which could be experienced as distressing, traumatic, or damaging. Thus, clinicians needed to achieve a careful balance, while also being sure to adapt to the changing needs of service users over time. Specialist skills, training and experience were seen as helpful to managing this balance. There were multiple accounts of service users valuing a framework for treatment in both individual and group contexts in which boundaries were straightforward and clear, but not too strict or judgmental.

Exclusion or discharge from services to enforce rules was viewed as punitive and could lead to feelings of rejection and abandonment and, consequently, to a deterioration in mental state.

Ending a therapeutic relationship due to a change in clinical staff or the service user being transferred between or discharged from services, could also leave service users feeling abandoned or rejected. Service users advocated gradual change in their support teams, with careful and planned handovers. Lack of choice of clinicians was also a recurring theme, with some people feeling they were allocated to clinicians with whom they found it difficult to establish strong relationships, or whom they felt to have negative attitudes, especially in generic services. Having a voice in care planning meetings was also identified in several studies as important for good quality care.

**Family and friends.** Service users considered service engagement with family and other key supporters another important aspect of support, since interpersonal relationships can provide emotional and practical support for service users to manage emotions and symptoms. Several sources described this as supporting recovery by allowing relatives to better understand CEN their needs and by improving communication and trust. Even in specialist settings, there were accounts of service users reporting that there was little provision of support (including mutual support as in carers' groups) and psychoeducation for carers.

**Peer support.** Peer relationships were identified in many papers as valuable for recovery, both in therapeutic groups and in more informal settings. Their value included fostering a sense of belonging and relieving loneliness. Service users could share experiences and support one another, for example when managing symptoms such as self-harm. However, service users accounts of peer support were limited to those provided as part of clinician-led group treatment, and did not include descriptions of services employing peer support workers with experience of CEN or establishing peer support schemes of any time.

**Group treatment.** Positive experiences of group treatment were often described, especially due to the feelings of belonging and acceptance that could be fostered. A challenging aspect, however, was achieving a balance between the giving and receiving of support, and providing clear structures for doing so. Service users could feel sometimes overwhelmed by the needs of others, or that their own needs had not been met, and good structure for managing this were appreciated. For example, some papers described allocating of time to each participant in a group as a helpful means of facilitating sharing and ensuring everyone can contribute. Although service users appreciated groups in which members were encouraged to talk openly, groups could also be emotionally draining at times, for example when topics such as self-harm and suicide were discussed. A challenge was to establish what could be brought to the group and fostering a culture of safety, whilst avoiding the potentially limiting and frustrating effects of unduly strict rules and boundaries. For example, there were accounts of service users not feeling comfortable with revealing self-harm because of the potential repercussions for doing so. Furthermore, problematic (including aggressive) within-group relationships between participants could at times emerge, leaving some service users feeling excluded by the group or choosing to exclude themselves. It was important for group leaders to carefully monitor group dynamics and intervene as needed.

## e) Impacts of "personality disorder" diagnosis

Whilst our main aim was to understand service user views on what constitutes good care, many papers also included discussions of the pros and cons of receiving a diagnosis of "personality disorder", including its impacts on the care they received from services. Negative consequences appeared especially prominent outside specialist "personality disorder" services. These included being excluded from services and treatments because of the diagnosis and

being met with stigmatising or stereotypical attitudes amongst clinicians and society. There were reports that once labelled in this way, service users were no longer seen as unwell or distressed but as "difficult", and that some symptoms that they experienced, including psychotic symptoms, were no longer considered genuine. For some, the contested and uncertain nature of the diagnosis made it more difficult to feel in control of their condition because there were so many myths, misinformation, and derogatory attitudes, including amongst their clinicians. Such views include that "personality disorders" are untreatable, that self-harming and other behaviours are merely manipulations to gain attention, and that service users with the diagnosis are liars, attention-seeking, unreasonable or difficult, manipulative, and take resources from other patients. Such attitudes only compounded service users' feelings of isolation, marginalisation, abandonment, or rejection. Others felt the diagnosis pathologised the impacts of the abuses they had experienced throughout their lives, resulting further trauma and a sense of victimhood.

However, some positive effects on mental health care associated with a receiving a diagnosis of "personality disorder" were also described, especially regarding access to treatment and improved self-understanding. Some papers described service users finding that the diagnosis helped them to reflect on their own feelings and responses, and to engage in treatment. This was especially true if the diagnosis seemed to fit their experiences, and if it was contextualised with helpful information about the condition and treatment options. Where it was accompanied by access to potential helpful therapy, there were reports of the diagnosis offering a sense of validation and relief.

How service users were told about their diagnosis seemed to influence how they subsequently felt about it. Being given the diagnosis by a clinician who understood the condition, who had time for discussion, and who was optimistic about the effectiveness of treatment and the likelihood of recovery was more likely to result in a positive experience. Attempts by clinicians to avoid or sidestep a diagnosis of "personality disorder" were seen as counterproductive by some, inadvertently indicating clinicians' negative attitudes about the condition and, possibly, invalidating service users' hopes and understanding of themselves.

## Discussion

### Main findings and implications

We found a substantial literature (47 papers) published since 2003, from which a generally consistent set of themes regarding experiences of care emerged. Some overall points regarding implications for achieving good practice can be drawn based on these. Firstly, reports of good practice and helpful treatment, as far as available, seemed to be largely confined to periods of care by services specialising in care for people with a "personality disorder" diagnosis, such as the multidisciplinary teams established for this purpose in parts of England [2]. However, specialist care was often hard to access and time-limited in a way that does not fit with the long journey towards "recovery" described by service users with CEN. Service user accounts across many papers suggest that care pathways are needed that take into account the long timescales involved in living with CEN, and the many set-backs often experienced. Holistic support from empathetic professionals with a good understanding of CEN is needed even during periods when service users are not engaged in intensive therapies. Transitions between stages of care need to be smooth and well-understood by all. In the care of conditions such as psychosis and bipolar, models such as early intervention and assertive outreach services and recovery teams have been developed to meet a range of service user needs over a long timescale. Development and implementation of such models for people with CEN has been much more limited even though this group likewise have long-term and fluctuating needs in many areas of their lives.

Secondly, good relationships and skilled support from clinicians who convey hope regarding long—term improvement in CEN are seen as central throughout pathways through the mental health care system. Service users tend to value highly clinicians who have the right skills to create safe spaces in individual and group treatment and manage exploration of challenging topics such as self-harm and trauma. Across the included studies, clinicians with the necessary skills and values seemed to be mainly found in specialist services, with at times appalling descriptions of lack of understanding and hopefulness, and stigmatising attitudes and behaviour elsewhere in primary and secondary mental health care. Addressing this appears central to achieving good practice, whether by designing pathways so that people with CEN normally receive care throughout from people with some specialist understanding of their condition, and/or by large-scale programmes to improve attitudes towards and understanding of CEN across the healthcare system. Stepped care models, in which some service users receive CEN interventions within generic services, are often advocated as a way of meeting needs of people with CEN across the care system. The success of such models is likely to require attitudes to CEN in generic services to be much more positive than has tended to be described in the current review [73]. Given the centrality of therapeutic relationships in recovery, service users also often advocated for a choice of therapist, but this seemed to be offered relatively rarely. Peer support appears, from the studies, reviewed to be an area with considerable scope for innovation: the mutual acceptance and understanding and sense of belonging available from peers is experienced as very helpful and validating, especially as loneliness appears to be a core difficulty in CEN. There were few accounts of harnessing this potential beyond therapeutic groups, although it seems to be a significant potential component of good practice [74].

Thirdly, as with other longer-term mental health conditions, care needs to meets a range of psychological, social and physical needs. Yet, service users reported that support from specialist services often focused mostly on self-harm and emotional regulation, with people who did not feel ready to focus on these issues or who had other care priorities sometimes excluded from care. Thus, we suggest that achieving good practice should involve designing holistic services that offer not only specific therapies (which are often highly structured and focused) but also support people with social and practical difficulties, looking after physical health, managing substance use, and with managing relationships and reducing loneliness.

Finally, being given a "personality disorder" diagnosis can have profound effects on all aspects of service users' experiences with mental health care services. On the one hand, the diagnosis was sometimes described as helpful in contextualizing distressing symptoms, especially if it was communicated in a sensitive manner, and could allow service users access to specialist care. However, it was also clear that stigmatising attitudes held by clinicians could be detrimental to a service user's sense of self-worth and ultimately impede their recovery. Such stigma was most common in generic mental health services and primary care, and could lead to pessimism amongst clinicians about the prospects of recovery and consequently to service users being denied access to care and treatment. These findings indicate a need for improved training for clinicians outside specialist 'personality disorder' services.

Throughout the included studies, there appeared to be frequent mismatch between service users' clear assessments of their needs for long-term engagement with holistic care delivered by empathetic clinicians with a realistic but hopeful understanding of CEN, and the service contacts they experienced. This reinforces the need to include people with lived experience of CEN and of using services, as well as their families and friends, in the development and assessment of services and care pathways. The perspectives of clinicians, investigated in an accompanying review [75], as well as those of family and friends, also need to be understood to ensure plans to improve care are feasible, and to develop approaches to reducing stigma and improving understanding and attitudes. Furthermore, we note also that trials of therapeutic interventions to date

have tended to focus on testing effects of specific therapies over relatively short durations [76]. The results of this study make clear that service user views of good practice tend more to focus on access to a broad and individualised treatment that lasts over the long term.

## Limitations

The process of concisely synthesising findings across many qualitative studies from different dates and countries inevitably leads to some loss of nuance and simplification of findings, while allowing cross-validation between studies regarding themes that are recurrent in different populations. There was considerable heterogeneity regarding participant characteristics, treatment type, and methods, but neither the reporting of data in most papers nor our approach of looking for commonalities between papers allowed us to identify differences by groups. Papers varied in inclusion criteria, with some samples primarily of people with "emotionally unstable" or "borderline personality disorder" diagnosis, others of mixed samples. However, even the latter seemed primarily focused on the difficulties of emotional regulation and impulsive behaviour that may lead to a "borderline personality disorder" diagnosis. Generalisability to their experiences of people with other "personality disorder" diagnoses is thus limited. Further, we planned to include studies that did not describe their sample as having a 'personality disorder' but were nevertheless considered by a senior psychiatrist in the research team to fit the diagnosis. We considered this to be important for the reasons described in our introduction and inclusion criteria. However, as shown in Table 2, almost all of the included papers specified that the included samples had received a 'personality disorder' diagnosis, self-identified as having the diagnosis, or were service users of specialist "personality disorder" services. This makes our work more clearly congruent with other research in this area, but does mean that our process may not have captured papers investigating a similar population in which the diagnosis was not explicit.

To exclude papers written regarding service systems very different from current ones, we only identified papers written since 2003, the year "Personality Disorder: No Longer A Diagnosis Of Exclusion" was published [9], so that potentially important experiences before that date will have been missed. Although eligible for inclusion, we did not find papers written in languages other than English. Most of the papers were from the UK, so that our results, as well as the perspective of the authors, may result in a disproportionate focus on the UK. However, we found that similar experiences tended to be reported in the various included countries as well as across the timespan of our study. However, except for China, all included countries were higher income, and most samples were either largely White or did not report ethnic background. Thus we could not assess the relevance of our findings to these wider global and societal contexts. This a key task for further research, especially given preliminary evidence that attitudes, help-seeking, patterns of difficulties and identity in this clinical group may be shaped in important ways by cultural context [77].

Participants in the included studies were mainly people who were to some extent engaged with services. Thus, the experiences of potentially the most dissatisfied and underserved group, people who are not engaged with any kind of care, are likely to be under-represented. We did not include in our searches the perspectives of family members and friends who support service users: there is evidence that the burden they experience may be substantial, but that they often report exclusion from decision making and support: capturing their perspectives in the qualitative literature will thus also be important in further research [78].

## Lived experience commentaries

The importance of providing individualised and holistic care instead of a "one-size-fits-all" approach in community services for people with CEN cannot be emphasised enough. We are

not defined by a "Personality Disorder" label, but should be respected and treated as the unique human beings that we are.

It is evident that the stigma of the diagnosis is still insidious—especially amongst staff in generic community mental health services. This is extremely disappointing to see 17 years after "Personality Disorders" were officially declared "No Longer a Diagnosis of Exclusion", which may be indicative of a culture resistant to change.

Unfortunately, it strongly resonates with some of our own lived experience that having to work with clinicians outside specialist services who demonstrate no real understanding of or empathy and respect for people with CEN often does more (iatrogenic) harm than good. In fact, hardly anything could be more re-traumatising than blatant 'malignant alienation', [79] which in any other context would be considered unthinkable.

In order to tackle such entrenched attitudes, we need a culture shift across community services. Mandatory CEN-specific training for clinicians should be co-produced with service users and embed helpful features of specialist services as well as trauma-informed care. However, any learning can only be successfully implemented in practice if it is consistently reinforced through role-modelling.

Overall, this meta-synthesis highlights a desperate need for change in order to provide the right care at the right time in more inclusive mental health services for people with CEN. Parity of esteem between services for CEN and other SMIs—where pathways are much better established and the importance of long-term support is widely recognised–is long overdue. Ultimately, we cannot afford to waste another 17 years without genuine progress towards treating people with CEN with the dignity and respect that they deserve.

Eva Broeckelmann and Jessica Russell

Two decades of research tell us that interventions need to float an individual's boat. The boats are ideally equipped for a long voyage, sail at their own pace, choose their destination and have kind, skilled staff on board.

This isn't big, clever or new. New research concurs with research written two decades ago. Why aren't we fixing those boats?

Evaluations demonstrated a positive change in negative attitudes and stigma amongst staff after attending co-designed and co-delivered KUF (Knowledge & Understanding Framework) training. Funding has since been cut.

Survivor led organisation Emergence CIC developed innovative ways of working that were co-delivered or led by survivors. The lack of inclusion of co-produced work within the review demonstrates a missing literature base. Survivor knowledge has been decimated alongside funding cuts.

The question isn't about what to do, or how to do it. The question is why aren't we?

Why keep sabotaging the boats we already know we need?

Tamar Jeynes

## Supporting information

**S1 Checklist. PRISMA 2009 checklist.**
(DOC)

**S1 File.**
(PDF)

## Author Contributions

**Conceptualization:** Chris Cooper, Sian Oram, Sonia Johnson.

**Data curation:** Luke Sheridan Rains, Athena Echave, Thomas Steare, Chris Cooper.

**Formal analysis:** Luke Sheridan Rains, Jessica Rees, Hannah Rachel Scott, Billie Lever Taylor, Eva Broeckelmann, Sarah Rowe.

**Methodology:** Luke Sheridan Rains, Athena Echave, Chris Cooper, Sian Oram, Sarah Rowe, Sonia Johnson.

**Supervision:** Luke Sheridan Rains, Sian Oram, Sonia Johnson.

**Writing – original draft:** Luke Sheridan Rains, Jessica Rees, Hannah Rachel Scott, Eva Broeckelmann, Phoebe Barnett, Tamar Jeynes, Jessica Russell, Sonia Johnson.

**Writing – review & editing:** Luke Sheridan Rains, Athena Echave, Jessica Rees, Hannah Rachel Scott, Billie Lever Taylor, Eva Broeckelmann, Thomas Steare, Phoebe Barnett, Chris Cooper, Tamar Jeynes, Jessica Russell, Sian Oram, Sarah Rowe, Sonia Johnson.

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
