## [Decision Letter · Decision Letter 0]

8 Dec 2020

PONE-D-20-34071

Service user experiences of community services for Complex Emotional Needs: A qualitative thematic synthesis

PLOS ONE

Dear Dr. Rains,

Thank you for submitting your manuscript to PLOS ONE. After careful consideration, we feel that it has merit but does not fully meet PLOS ONE’s publication criteria as it currently stands. Therefore, we invite you to submit a revised version of the manuscript that addresses the points raised during the review process.

Please consider the reviewer points. 

We look forward to receiving your revised manuscript.

Kind regards,

Andrew Soundy

Academic Editor

PLOS ONE

Journal Requirements:

2. Please include your tables as part of your main manuscript and remove the individual files. Please note that supplementary tables should be uploaded as separate "supporting information" files.

"This paper presents independent research commissioned and funded by the National Institute

for Health Research (NIHR) Policy Research Programme, conducted by the NIHR Policy Research Unit

(PRU) in Mental Health. The views expressed are those of the authors and not necessarily those of the

NIHR, the Department of Health and Social Care or its arm’s 591 length bodies, or other government

departments."

Reviewers' comments:

Reviewer's Responses to Questions

**Comments to the Author**

1. Is the manuscript technically sound, and do the data support the conclusions?

Reviewer #1: Yes

Reviewer #2: Yes

2. Has the statistical analysis been performed appropriately and rigorously? 

Reviewer #1: N/A

Reviewer #2: N/A

3. Have the authors made all data underlying the findings in their manuscript fully available?

Reviewer #1: Yes

Reviewer #2: Yes

4. Is the manuscript presented in an intelligible fashion and written in standard English?

Reviewer #1: Yes

Reviewer #2: Yes

5. Review Comments to the Author

Reviewer #1: Thank you for the opportunity to review this timely and important manuscript. The paper reviewed qualitative papers pertaining to the service user experience of using community mental health services in people with experience of personality disorder, or Complex Emotional Needs (CEN). The paper was generally well written and coherent, however there were a few typographical errors or unfinished sentences.

Specific minor comments:

Abstract:

• I'm interested in why only examples of 'good care' was included in the review. Why were bad examples of care not included as this presents an opportunity to learn from less effective models. Also, how were experiences of good and bad care differentiated?

Background:

• "Personality Disorder" and Complex Emotional Needs is being used interchangeably. Whilst this is understandable, given the debates about diagnosis, as mentioned on page 5 lines 93-101. It may be, however, difficult for a reader who is not well versed in these debates to follow the text. Perhaps it may be worthwhile to move the text from page 5 lines 93-101 to earlier in the background section, to set the scene for the reader.

Methods:

• It is reported that studies that describe repeated self-harm, suicide attempts, complex trauma or complex PTSD and emotion dysregulation and instability were also included on a case by case basis. What were the criteria that was used to determine whether this met the criteria for this study?

• Were only studies published in English included in the review or were other languages also eligible? Whilst this is explained in the discussion, clarification in the method section would be helpful to the reader.

Results:

• The results of the thematic synthesis is well written and structured in a way that is easy to understand to the reader. To strengthen the synthesis, an indication of the proportion of studies (and which studies) reported on each of the themes and sub-themes would have been helpful to discern the strength of the themes and sub-themes.

• The identification of positive approaches to care is important. However, I wondered whether the phrase 'positive approaches' is the best term, as some readers may confuse this with positive psychotherapy.

Discussion:

• The inclusion of a lived experience commentary from three individuals with lived experience was insightful and helped boost the reliability of the findings. However, were the people with lived experience provided with the findings of the paper when writing their contributions and were there any comments that were made during the process, which altered the manner in which findings were presented?

General:

• Line 358: repetition of the same idea

• Line 374: Sentence not finished

• Lines 599, 601: references need editing

Reviewer #2: Complex emotional needs synthesis

Overall, a very interesting and worthy paper reporting a qualitative synthesis of the experiences of service users with complex emotional needs (i.e. personality disorders) with community health services

Introduction: lines 50-51: clarify what type of professionals you refer to

Introduction: line 58. You provide a rationale for the use of the term complex emotional needs instead of ‘personality disorder’ later on in the introduction. I would suggest using the term ‘personality disorder’ consistently up until this point because it is unclear whether complex emotional needs is used synonymously with personality disorder until this rationale.

Introduction: lines 65-67. Contextualise the importance of involving service user and carer/family perspectives in service design. This is part of a broader shift in public policy and mental health system development, not limited to the field of complex emotional needs/personality disorders’.

Methods: lines 129-132 who performed the full text screening including double screening and discussion with senior reviewers? Suggest putting the author/researchers initials in brackets after each research activity

Quality assessment and data analysis: 156-157 also indicate the two researchers who performed the quality assessment. The same applies for data analysis/thematic analysis.

Results: line 178-188 reference the relevant papers for each of the sample types e.g. “28 papers reported data from people diagnosed with “Borderline Personality Disorder” (REFERENCES X-X). Although this information is included in the Table 1, it is not ordered as such so it would be useful for the reader to know this from the text. I would suggest the same for the description of the quality assessment domains.

Results: line 236 clarify what you mean by recovery. Does this refer to the concept of personal recovery or clinical recovery or both?

Results: lines 266-269 review this sentence for meaning

Results: lines 356-361 review these sentences for meaning

Results: line 374 ‘the value of peer support’ is an unfinished sentence

Discussion: it would be helpful to clarify what type of professionals (and their training/background) provide ‘specialist care’, given that a key finding is that the quality of care was largely better in these settings.

Discussion: discuss the implications of focusing solely on the perspectives on consumers and not carers, family members and other supporters who often have a central role to play in the support and care of people with complex emotional needs

Discussion: the synthesized literature is mainly from English-speaking countries and all was from high-income countries with the exception of China. Discuss how this shaped the centralisation of the individual in the finding and implications arising from the papers. For example, would you expect these findings to hold in contexts/for people who place less emphasis on the individual consumer and more on the collective sense of self. I’m thinking particularly black and ethnic minority groups in the UK, culturally and linguistically diverse groups in Australia, etc

6. PLOS authors have the option to publish the peer review history of their article (what does this mean?). If published, this will include your full peer review and any attached files.

Reviewer #1: No

Reviewer #2: **Yes: **Teresa Hall

---

## [Author Response · Author response to Decision Letter 0]

28 Jan 2021

Editor comments:

1. Comment: Please ensure that your manuscript meets PLOS ONE's style requirements, including those for file naming. 

Reply: We believe that our manuscript is now formatted according to PLOS ONE’s style requirements. Specifically, we have added S* text (p.7 line 131; p.9 line 163; p.25 line 216) to cite supplementary materials. Secondly, we have added in the corresponding author details to p.1 (line 18). Thirdly, we have changed the formatting of the headings so that they are all bolded and font size 18. 

2. Comment: Please include your tables as part of your main manuscript and remove the individual files. Please note that supplementary tables should be uploaded as separate "supporting information" files.

Reply: We have now added the tables to the manuscript. Table 1 is on p.12 to p.21. Table 2 is on p.23 to 24. Table 3 is p. 26 to 29.

3. Comment: Thank you for stating the following in the Funding Section of your manuscript: "This paper presents independent research commissioned and funded by the National Institute for Health Research (NIHR) Policy Research Programme, conducted by the NIHR Policy Research Unit (PRU) in Mental Health. The views expressed are those of the authors and not necessarily those of the NIHR, the Department of Health and Social Care or its arm’s 591 length bodies, or other government departments." We note that you have provided funding information that is not currently declared in your Funding Statement. However, funding information should not appear in the Acknowledgments section or other areas of your manuscript. We will only publish funding information present in the Funding Statement section of the online submission form. Please remove any funding-related text from the manuscript and let us know how you would like to update your Funding Statement. Currently, your Funding Statement reads as follows: "The funders had no role in study design, data collection and analysis, decision to publish, or preparation of the manuscript." Please include your amended statements within your cover letter; we will change the online submission form on your behalf.

Reply: We have removed the funding and acknowledgement statements from the manuscript. We would like to change the funding statement in our online submission to: "This paper presents independent research commissioned and funded by the National Institute for Health Research (NIHR) Policy Research Programme, conducted by the NIHR Policy Research Unit (PRU) in Mental Health. The views expressed are those of the authors and not necessarily those of the NIHR, the Department of Health and Social Care or its arm’s 591 length bodies, or other government departments."

4. Comment: We note that you have indicated that data from this study are available upon request. PLOS only allows data to be available upon request if there are legal or ethical restrictions on sharing data publicly. For information on unacceptable data access restrictions, please see http://journals.plos.org/plosone/s/data-availability#loc-unacceptable-data-access-restrictions. In your revised cover letter, please address the following prompts: a) If there are ethical or legal restrictions on sharing a de-identified data set, please explain them in detail (e.g., data contain potentially identifying or sensitive patient information) and who has imposed them (e.g., an ethics committee). Please also provide contact information for a data access committee, ethics committee, or other institutional body to which data requests may be sent. b) If there are no restrictions, please upload the minimal anonymized data set necessary to replicate your study findings as either Supporting Information files or to a stable, public repository and provide us with the relevant URLs, DOIs, or accession numbers. Please see http://www.bmj.com/content/340/bmj.c181.long for guidelines on how to de-identify and prepare clinical data for publication. For a list of acceptable repositories, please see http://journals.plos.org/plosone/s/data-availability#loc-recommended-repositories. We will update your Data Availability statement on your behalf to reflect the information you provide.

Reply: Since this is a review of published data, the data are publicly available through the included published papers. The statement we made regarding data availability referred to our data analysis coding framework rather than data per se. Sufficient information is available regarding the included papers for replication of our study. 

5. Comment: Please include captions for your Supporting Information files at the end of your manuscript, and update any in-text citations to match accordingly. Please see our Supporting Information guidelines for more information: http://journals.plos.org/plosone/s/supporting-information.

Reply: We have now added captions to the end of the manuscript. We have added in-text citations where needed: p.7 line 131; p.9 line 163; p.25 line 216. 

Reviewer #1 comments:

Thank you for the opportunity to review this timely and important manuscript. The paper reviewed qualitative papers pertaining to the service user experience of using community mental health services in people with experience of personality disorder, or Complex Emotional Needs (CEN). The paper was generally well written and coherent, however there were a few typographical errors or unfinished sentences.

Specific minor comments:

Abstract:

Comment: • I'm interested in why only examples of 'good care' was included in the review. Why were bad examples of care not included as this presents an opportunity to learn from less effective models. Also, how were experiences of good and bad care differentiated?

Reply: Our aim was to provide a foundation for future service development and research by identifying good practice. However, as the reviewer rightly suggests, the papers we included contain a great deal of material on what is poor or problematic practice in services. We draw on this throughout our results when describing the issues service users face can face when receiving care. As such, while our aim is to describe good practice, to do so meant that we drew explicitly or implicitly on all the relevant material in the studies. However, we agree entirely that this is an important clarification to make in our manuscript, which we have now done in the Quality Assessment and Analysis section in the Methods (line 173). 

Background:

Comment: • "Personality Disorder" and Complex Emotional Needs is being used interchangeably. Whilst this is understandable, given the debates about diagnosis, as mentioned on page 5 lines 93-101. It may be, however, difficult for a reader who is not well versed in these debates to follow the text. Perhaps it may be worthwhile to move the text from page 5 lines 93-101 to earlier in the background section, to set the scene for the reader.

Reply: Thank you for your comment and we agree that this would help to improve clarity for the reader. We have now moved this section to the second paragraph of the background section – line 56 onwards. 

Methods:

Comment: • It is reported that studies that describe repeated self-harm, suicide attempts, complex trauma or complex PTSD and emotion dysregulation and instability were also included on a case by case basis. What were the criteria that was used to determine whether this met the criteria for this study?

Reply: The research team included a senior academic psychiatrist who considered whether a primary diagnosis of CEN for at least 50% of the population was likely to be appropriate given the description of the sample. We did not include papers that did not meet this requirement, or if they did not provide enough information to make this judgement. 

Comment: • Were only studies published in English included in the review or were other languages also eligible? Whilst this is explained in the discussion, clarification in the method section would be helpful to the reader.

Reply: We included papers from other languages as well. However, as noted in the limitations section, we did not find papers written in languages other than English. We state in the methods section (line 139) that ‘No limits were placed on the language or location of publications.’ 

Results:

Comment: • The results of the thematic synthesis is well written and structured in a way that is easy to understand to the reader. To strengthen the synthesis, an indication of the proportion of studies (and which studies) reported on each of the themes and sub-themes would have been helpful to discern the strength of the themes and sub-themes.

Reply: We agree that this is a helpful addition to the manuscript, and we have now added references to the papers supporting each theme to table 3. 

Comment • The identification of positive approaches to care is important. However, I wondered whether the phrase 'positive approaches' is the best term, as some readers may confuse this with positive psychotherapy.

Reply: We have now changed this to ‘helpful approaches to care’ throughout the manuscript. 

Discussion:

Comment: • The inclusion of a lived experience commentary from three individuals with lived experience was insightful and helped boost the reliability of the findings. However, were the people with lived experience provided with the findings of the paper when writing their contributions and were there any comments that were made during the process, which altered the manner in which findings were presented?

Reply: Thank you for your comments. The lived experience researchers did have the manuscript prior to writing their commentaries. The commentaries are written by the LE researchers independently of the other authors, and the LE commentaries are their own work. In terms of their other contributions to the paper: one LE researcher (EB) was part of the group who performed the data analysis. All LE researchers contributed to writing the discussion alongside three of the other authors (SJ, SR, LSR); along with all other the authors, they also contributed to editing and improving the drafted manuscript and approved it prior to submission. 

Comment: General:

• Line 358: repetition of the same idea

• Line 374: Sentence not finished

• Lines 599, 601: references need editing

Reply: Thank you for raising these points. We have now addressed them. The first point (in the ‘Relationship dynamics and involvement’ section in the Results) has been addressed by removing ‘their treatment, such as’. The second point, which is in the same section, has been addressed by removing the incomplete sentence. The third point regarding has been addressed and the references have now been edited. 

Reviewer #2: Complex emotional needs synthesis

Overall, a very interesting and worthy paper reporting a qualitative synthesis of the experiences of service users with complex emotional needs (i.e. personality disorders) with community health services

Comment: Introduction: lines 50-51: clarify what type of professionals you refer to

Reply: Thank you for your comment. We have now clarified this by making it clear that we are referring to ‘clinical professionals in both health and mental health settings’ (line 53). 

Comment: Introduction: line 58. You provide a rationale for the use of the term complex emotional needs instead of ‘personality disorder’ later on in the introduction. I would suggest using the term ‘personality disorder’ consistently up until this point because it is unclear whether complex emotional needs is used synonymously with personality disorder until this rationale.

Reply: Thank you for your comment and we agree that changing how we use these terms would help to improve clarity for the reader. We have now moved our section discussing terminology to the second paragraph of the background section, line 56 onwards, so that we can more easily use complex emotional needs throughout. 

Comment: Introduction: lines 65-67. Contextualise the importance of involving service user and carer/family perspectives in service design. This is part of a broader shift in public policy and mental health system development, not limited to the field of complex emotional needs/personality disorders’.

Reply: Thank you for the suggestion and we agree that this would improve the manuscript. We have now reworked this section (line 84 onwards) and included some references to relevant literature.

Comment: Methods: lines 129-132 who performed the full text screening including double screening and discussion with senior reviewers? Suggest putting the author/researchers initials in brackets after each research activity

Reply: Thank you for your helpful suggestion. We have now included initials of the researchers involved in screening (line 135) and senior researchers (line 137). We have not included the initials of the researchers responsible for the first stage of screening, employed for the wider programme of work, as they are no longer in academia and opted not to be authors on the paper.

Comment: Quality assessment and data analysis: 156-157 also indicate the two researchers who performed the quality assessment. The same applies for data analysis/thematic analysis.

Reply: We have now included initials of the researchers involved (line 167).

Comment: Results: line 178-188 reference the relevant papers for each of the sample types e.g. “28 papers reported data from people diagnosed with “Borderline Personality Disorder” (REFERENCES X-X). Although this information is included in the Table 1, it is not ordered as such so it would be useful for the reader to know this from the text. I would suggest the same for the description of the quality assessment domains.

Reply: Thank you for this suggestion, and while we agree that in many situations such a change could make it easier for the reader to identify particular papers, we also think that including this information in our manuscript may make it overly cumbersome for the reader because there are 47 included papers. We would need to include 47 references in the section on diagnoses and 49 in the quality appraisal section. As such, on balance, we feel that we would prefer not to add this information to the text as it is presented in tables 1 and 2. However, we are happy to revisit this issue and discuss further.

Comment: Results: line 236 clarify what you mean by recovery. Does this refer to the concept of personal recovery or clinical recovery or both?

Reply: We agree that this is a helpful improvement and we have now clarified this point to make clear that we mean both (line 253).

Comment: Results: lines 266-269 review this sentence for meaning

Results: lines 356-361 review these sentences for meaning

Results: line 374 ‘the value of peer support’ is an unfinished sentence

Reply: Thank you for highlighting these issues and we have now addressed them. The first sentence (line 283) now reads: “There were reports that having boundaries and consequences for self-harm can be helpful…” The second sentence (in Relationship Dynamics and involvement) now reads ‘There were accounts of service users wanting to be challenged by their clinician and pushed to progress in treatment, for example, to stop self-harming.’ We have now removed the unfinished sentence from the ‘Relationship Dynamics and Involvement’ section.

Comment: Discussion: it would be helpful to clarify what type of professionals (and their training/background) provide ‘specialist care’, given that a key finding is that the quality of care was largely better in these settings.

Reply: We agree that it is helpful to clarify this issue and we have now added more to the discussion (line 475). 

Comment: Discussion: discuss the implications of focusing solely on the perspectives on consumers and not carers, family members and other supporters who often have a central role to play in the support and care of people with complex emotional needs

Reply: This is an important point and we have now added a longer discussion to our limitations section (line 557 onwards). 

Comment: Discussion: the synthesized literature is mainly from English-speaking countries and all was from high-income countries with the exception of China. Discuss how this shaped the centralisation of the individual in the finding and implications arising from the papers. For example, would you expect these findings to hold in contexts/for people who place less emphasis on the individual consumer and more on the collective sense of self. I’m thinking particularly black and ethnic minority groups in the UK, culturally and linguistically diverse groups in Australia, etc

Reply: This is another important point and we have now added more discussion of this issue in our limitations section (line 549 onwards).

---

## [Decision Letter · Decision Letter 1]

24 Feb 2021

Service user experiences of community services for Complex Emotional Needs: A qualitative thematic synthesis

PONE-D-20-34071R1

Dear Dr. Rains,

We’re pleased to inform you that your manuscript has been judged scientifically suitable for publication and will be formally accepted for publication once it meets all outstanding technical requirements.

Kind regards,

Andrew Soundy

Academic Editor

PLOS ONE

Additional Editor Comments (optional):

Reviewers' comments:

Reviewer's Responses to Questions

**Comments to the Author**

1. If the authors have adequately addressed your comments raised in a previous round of review and you feel that this manuscript is now acceptable for publication, you may indicate that here to bypass the “Comments to the Author” section, enter your conflict of interest statement in the “Confidential to Editor” section, and submit your "Accept" recommendation.

Reviewer #1: All comments have been addressed

Reviewer #2: All comments have been addressed

2. Is the manuscript technically sound, and do the data support the conclusions?

Reviewer #1: Yes

Reviewer #2: Yes

3. Has the statistical analysis been performed appropriately and rigorously? 

Reviewer #1: N/A

Reviewer #2: N/A

4. Have the authors made all data underlying the findings in their manuscript fully available?

Reviewer #1: Yes

Reviewer #2: Yes

5. Is the manuscript presented in an intelligible fashion and written in standard English?

Reviewer #1: Yes

Reviewer #2: Yes

6. Review Comments to the Author

Reviewer #1: Thank you for considering my comments. My comments have been sufficiently addressed and I believe that this paper is suitable for publication.

Reviewer #2: The authors have made a satisfactory response to my original review. The ony query I have is that I still believe the results section would be strengthened by referencing the key findings against the original studies. However, I accept the authors' justification as to why this is not necessary.

7. PLOS authors have the option to publish the peer review history of their article (what does this mean?). If published, this will include your full peer review and any attached files.

Reviewer #1: No

Reviewer #2: **Yes: **Teresa Hall

---

## [Editor Report · Acceptance letter]

12 Apr 2021

PONE-D-20-34071R1 

Service user experiences of community services for Complex Emotional Needs: A qualitative thematic synthesis 

Dear Dr. Sheridan Rains:

I'm pleased to inform you that your manuscript has been deemed suitable for publication in PLOS ONE. Congratulations! Your manuscript is now with our production department. 

Kind regards, 

on behalf of

Dr. Andrew Soundy 

Academic Editor

PLOS ONE